# Blood-Transfusion Risk Factors after Intramedullary Nailing for Extracapsular Femoral Neck Fracture in Elderly Patients

**DOI:** 10.3390/jfmk8010027

**Published:** 2023-02-20

**Authors:** Gianluca Testa, Marco Montemagno, Andrea Vescio, Giuseppe Micali, Rosario Perrotta, Francesco Lacarrubba, Teresio Avitabile, Guido Basile, Vito Pavone

**Affiliations:** 1Department of General Surgery and Medical Surgical Specialties, Section of Orthopaedics and Traumatology, University Hospital Policlinico-San Marco, University of Catania, 95123 Catania, Italy; 2Department of General Surgery and Medical Surgical Specialties, Dermatology Clinic, University Hospital Policlinico-San Marco, University of Catania, 95123 Catania, Italy; 3Department of General Surgery and Medical Surgical Specialties, Section of Plastic and Reconstructive Surgery, University of Catania, 95123 Catania, Italy; 4Department of General Surgery and Medical Surgical Specialties, Section of Ophthalmology, University Hospital Policlinico-San Marco, University of Catania, 95123 Catania, Italy; 5Department of General Surgery and Medical Surgical Specialties, Section of General Surgery, University Hospital Policlinico-San Marco, University of Catania, 95123 Catania, Italy

**Keywords:** blood loss, intertrochanteric fracture, pertrochanteric fracture, hip fracture, complication, ASA score, Charlson Comorbidity Index, risk factor, predictor factor

## Abstract

Background: Extracapsular femoral neck fractures (eFNF) are the third most common type of fracture in traumatology. Intramedullary nailing (IMN) is one of the most frequently used ortho-pedic treatments for eFNF. Blood loss is one of the main complications of this treatment. This study aimed to identify and evaluate the perioperative risk factors that lead to blood transfusion in frail patients with eFNF who undergo IMN. Methods: From July 2020 to December 2020, 170 eFNF-affected patients who were treated with IMN were enrolled and divided into two groups according to blood transfusion: NBT (71 patients who did not need a blood transfusion), and BT (72 patients who needed blood transfusion). Gender, age, BMI, pre-operative hemoglobin levels, in-ternational normalized ratio (INR) level, number of blood units transfused, length of hospital stay, surgery duration, type of anesthesia, pre-operative ASA score, Charlson Comorbidity Index, and mortality rate were assessed. Results: Cohorts differed only for pre-operatively Hb and surgery time (*p* < 0.05). Conclusion: Patients who have a lower preoperative Hb level and longer surgery time have a high blood-transfusion risk and should be closely followed peri-operatively.

## 1. Introduction

Extracapsular femoral neck fractures (eFNFs) are the third most common type of fracture in traumatology [1]. They mainly occur in older patients, resulting in significant morbidity and mortality. Population aging worldwide is rapidly accelerating, from 461 million people aged over 65 years in 2004 to an estimated 2 billion people by 2050. Therefore, eFNFs will increase over the next decades [2].

In the literature, the rise in intracapsular fractures is well described through the average age of presentation with proximal femoral fractures and is associated with persistently high mortality (33%) and morbidity, which are greater than in patients with eFNFs [3].

Some authors assumed that there is an increasing incidence of proximal femoral fractures in the population, which rises by about 2 to 3 times above the age of 50 years old. These fractures especially affect females [4,5].

The epidemiologic studies by Lamb et al. [6] confirmed that the relative incidence of proximal femoral fractures is higher in elderly females, with reduced bone-mineral density and reduced trochanteric bone strength as the main causes of fractures.

Intramedullary nailing (IMN) is a common orthopedic treatment for eFNFs, and it is recommended by the American Academy of Orthopaedic Surgeons (AAOS) for stable and unstable intertrochanteric fractures [7,8]. Surgical treatment allows early patient mobilization, decreases the risk of complications, and improves patient outcomes [9,10,11,12]. Several factors, including osteoporosis, malnutrition, decreased physical activity, impaired vision, neurologic disease, poor balance, muscle atrophy, and bedsores, could be associated with older people [10,11,13].

In the literature, numerous clinical features are associated with post-operative outcomes, such as blood transfusions, which have a predominant role [14,15,16,17]. Femoral-neck-fracture patients tend to be frailer; they present with acute hemorrhage from the fracture and they are often dehydrated or hypovolemic on admission [18]. A recent Korean nationwide representative cohort study showed that blood transfusion was required in 75% of femoral-neck-fracture patients, and in the UK, blood loss requiring transfusion is the most common eFNF after chest infection [19,20].

The hemoglobin levels in these kinds of patient were found to drop significantly in the first 24 h of hospitalization; this should be taken into consideration when planning transfusion [21]. Compared with patients with intracapsular FNF, patients affected by eFNFs have more blood loss and a higher blood-transfusion rate [22].

A relevant negative factor in eFNF is the concept of hidden blood loss (HBL), which was widely investigated in the recent literature [22,23,24]. Occurring before the operation and on the first postoperative days, it is caused by partial injury before admission, anti-coagulant treatment after surgery or insufficient hemostasis.

Recent studies related to hip-fracture patients found that receiving a blood transfusion during an admission led to an increased risk of one-year mortality of almost two-and-a-half times [25]; in addition, the effect of postoperative transfusion was found to be a common occurrence in geriatric-fragility eFNF, with a higher significant risk of 30-day mortality, hospital-readmission rate and length of stay [26].

The aim of this study was to identify and evaluate the perioperative risk factors that led to blood transfusion in frail patients with eFNF who underwent IMN.

## 2. Materials and Methods

### 2.1. Study Design

The study was conducted according to the Strengthening the Reporting of Observational Studies in Epidemiology (STROBE) Statement’s guidelines for reporting observational studies [27]. This prospective cross-sectional study was approved by the ethics committee at our institution (reference number 160/2020/PO).

### 2.2. Sample

From July 2020 to December 2021, 170 FNF-affected patients who were surgically treated with IMN were enrolled.

The inclusion criteria were as follows: (1) confirmed diagnosis of eFNF, with a fracture pattern of AO/OTA 31.A1, 31.A2, 31.A3; (2) low-energy injury (e.g., falls from a standing height); (3) patient age over 65 years; and (4) IMN procedure. The exclusion criteria were as follows: (1) associated fractures; (2) pathological, peri-prosthetic, shaft, or distal femoral fractures; (3) patients under 65 years of age; and (4) open or pathological fracture (history of bone tumor, blood, or hemorrhagic disease, chronic liver disease).

All patients were admitted through the emergency department with the following demographic and clinical data captured by one author (MM): gender, age at the time of trauma, body-mass index (BMI), involved side, fracture type, comorbid diseases (e.g., hypertension and diabetes), pre- and post-operative hemoglobin (Hb) levels, international normalized ratio (INR) level, number of blood bags transfused, length of hospital stay, time to surgery, surgery duration, nail implant, type of anesthesia (general anesthesia (GA) or combined spinal–epidural anesthesia (CSEA)), American Society of Anesthesiologists (ASA) scores, and Charlson Comorbidity Index (CCI).

According to the blood-transfusion criteria described below, the patients were divided into NBT (patients who did not need a blood transfusion) and BT (patients who needed and received a blood transfusion) cohorts. All the patient data were collected and analyzed by two authors (MM and AV).

### 2.3. Surgical Technique, Intervention, and Blood-Transfusion Criteria

Open or closed eFNF reduction and internal fixation were achieved with the patient in the supine position on a radiolucent table with the injured leg draped freely or on a traction table. The lateral supra-trochanteric approach was used for each patient. All intramedullary nails were distally locked after the surgery.

The same expert surgical team performed all of the surgeries. All patients underwent initial postoperative radiographic imaging to confirm the quality of reduction and implant position. Cefazolin was used for prophylaxis of surgical-site infection. In cases of cefazolin or penicillin allergy, patients were administered clindamycin instead.

According to the Clinical Practice Guidelines From the American Association of Blood Banks (AABB): Red Blood Cell Transfusion Thresholds and Storage [28], patients were transfused only if they had a hemoglobin level lower than 8 g/dL with unstable vital signs or symptoms of severe anemia (heart rate >100 bpm, systolic blood pressure <90 mmHg, chest pain, asthenia).

### 2.4. Risk Factors and Outcome Measurements

Gender, age, BMI, pre-operative Hb levels, INR level, number of blood units transfused, length of hospital stay (days from the department admission to the discharge), time to surgery (days from Emergency Room (ER) admission to surgery), surgery duration (minutes from incision to suture), type of anesthesia (GA or CSEA), pre-operative ASA score, and CCI were considered to be comparison parameters between the NBT and BT cohorts. The relative risk (RR) and the number of blood bags transfused were calculated for each parameter for both groups. The cut-off values were as follows: patient’s gender (female sex), age (over 80 years), BMI (less than 25 kg/m^2^), Hb (<8 g/dL), INR (1.00), hospital stay (over 7 days), time to surgery (over 48 h), surgery duration (over 45 min), type of anesthesia (GE), ASA (over 2), and CCI (over 4).

### 2.5. Statistical Analysis

Continuous data are presented as the mean and standard deviation. The *t*-test was used to compare the mean age, BMI, Hb, INR, length of hospital stay, time to surgery, surgery duration, pre-operative ASA score, and CCI between the two groups. The Chi-square test was used to verify the homogeneity of the two groups based on gender and to compare the types of anesthesia. The RR was used to quantify the risk factor. Pearson’s ρ correlation coefficient was chosen to assess the correlation between parameters and the number of blood bags transfused. The selected threshold for statistical significance was *p* < 0.05. To evaluate survival rates and possible influential factors, Kaplan–Meier analysis was conducted on NBT vs. BT group with a 1.5-year follow-up (using the log-rank test for statistical significance). All statistical analyses were performed using SPSS version 24.0 statistical software (IBM Corp., Armonk, NY, USA).

## 3. Results

### 3.1. Patients

Among 170 patients with displaced AO/OTA 31.A1, 31.A2, and 31.A3 FNFs, 143 subjects (41 men (28.7%) and 102 women (71.3%)) were eligible and selected for the study. There were 71 male patients (24 (33.8%) and 47 female patients (66.2%) women) in Group NBT, with a mean age of 80.2 ± 10.7 years (range 65–94 years). There were 72 subjects (17 (23.6%) men and 55 (76.4%) women) in Group BT, with a mean age of 82.9 ± 7.6 years (range 65–99 years; Table 1). There were no differences in the patients’ de-mographic information (Table 1).

### 3.2. Group Comparisons

Table 1 shows no statistically significant differences between the cohorts for gender (*p* = 0.17), age (*p* = 0.07), BMI (*p* = 0.65), INR (*p* = 0.88), hospital stay (*p* = 0.44), time to surgery (*p* = 0.64), type of anesthesia (*p* = 0.65), ASA (*p* = 0.81), or CCI (*p* = 0.19). Pre-operatively, the Hb and surgery-duration parameters showed statistical differences (*p* < 0.001; Figure 1).

### 3.3. Relative-Risk Parameters

Higher RR values were found for the following parameters: female gender (1.43, 95% CI 0.8–2.4), age > 80 years (1.52, 95% CI 0.9–2.8), hospital stay over 7 days (1.16, 95% CI 0.45–2.8) and surgery duration over 45 min (1.37, 95% CI 0.80–2.3).

### 3.4. Cohort-Parameter Correlation

No significant correlations were found between the risk factors and the blood-transfusion needs (*p* > 0.05), except for the pre-operative Hb levels (ρ = −0.49, *p* < 0.001) and surgery duration (ρ = −0.28, *p* = 0.001) (Figure 2).

### 3.5. Survival Analysis

The one-year mortality rates resulted in 20.8% for the BT group and 19.7% for the NBT group. The probability of survival in the BT- and NBT-group patients dropped to about 50% over 153 days (95% C.I 127–236) and 190 days (95% C.I 164–326), respectively (Figure 3). The differences between the groups on the Kaplan–Meier analysis did were not statistically significant (χ^2^ = 0.987, *p* = 0.320) (Figure 3).

## 4. Discussion

Our data showed that the patients who underwent a blood transfusion had lower preoperative Hb levels and longer durations of surgical treatment. Moreover, BMI, INR, general anesthesia, ASA and CCI scores, surgery duration, and delays until surgery of more than 48 h were not found to be relevant parameters for requiring a blood transfusion. The main relative risk factors for red-blood-cell transfusion were female gender, age >80 years, and hospital stay >7 days, while the preoperative Hb levels and surgical durations were inversely and directly proportional, respectively, to the number of blood bags transfused.

The AAOS Clinical Practice Guideline: Management of Femoral Neck Fractures in the Elderly [8] recommends a blood-transfusion threshold of no higher than 8 g/dL in asymptomatic postoperative patients, and there is strong evidence supporting this recommendation. The need for blood transfusions was associated with the risk factors for early and one-year mortality in frail elderly eFNF patients [29]. Identifying blood-transfusion risk factors can theoretically reduce mortality. Our data confirmed this evidence even though no statistical differences in survival were found when comparing the one-year mortality between the BT patients and the NBT-group patients. Several trials [30,31] suggested that blood transfusions increase short-term functional outcomes and prevent delirium in the frail elderly population. However, blood transfusions were correlated with a downregulation in the immune response and an increase in the postoperative-infection rate, such as superficial-wound infection and urinary-tract infection [32,33], which can cause transfusion reactions (such as shivering and high fever), prolong the lengths of hospital stays, and increase the cost of treatment [34].

To determine the patients who were at high risk of perioperative anemia and to avoid unnecessary blood transfusions, several assessment parameters were proposed [15,17,35,36,37]. Lower preoperative Hb levels were found in patients who needed a blood transfusion compared to the other cohort. Previous researchers [38] showed that a preoperative Hb value lower than 120 g/L increases the risk of postoperative blood transfusion by five times in patients with femoral neck fractures. Moreover, a preoperative Hb level of less than 8 g/dL was strongly correlated with a higher number of blood-bag transfusions. These findings were confirmed by the statistical results of this prospective study; low pre-operative Hb levels (10.6 g/dL, on average) in elderly patients with eFNF undergoing IMN surgery were found to be a major risk factor for blood transfusion.

Elderly female patients were found to be the most vulnerable to the risks associated with blood transfusion. The reasons for this may be related to lower baseline hemoglobin levels in women compared to the male sex. These results are widely confirmed by the literature from recent years; Desai et al. [39], for instance, found that women were 1.54 times more likely to have a blood transfusion than men after a multivariate analysis.

Our study found no significant differences between the NBT and BT cohorts in terms of the influence of BMI on blood transfusion. These findings are in contrast to the recent literature, which suggests a relative risk of low BMI in these patients [26,40]; the authors of these studies stated that the protective effect of higher BMI on the risk of transfusion may be related to the overall increase in blood volume in the body. Obese patients may have a lower percentage of estimated blood volume lost during surgical procedure, compared to patients with a lower BMI, although they may have surgical problems, such as greater blood loss due to larger incisions, complex fracture reduction and longer surgical duration.

Regarding general or spinal anesthesia for patients who are undergoing femoral neck surgery and similar surgeries, the literature suggests avoiding hypotension and deliriant drugs [8,41]. General anesthesia is related to an increased probability of in-hospital mortality compared with regional anesthesia, and a risk of acute respiratory failure with increased length of hospital stay and readmission [42]. Patients who underwent GA reported a risk of blood transfusion that was four times greater than in those who underwent CSEA, and comparable risk rates were reported in intracapsular and extracapsular fractures [17,36]. Basques et al. [43] suggested that there is a difference in hemodynamics between GA and spinal anesthesia. Patients with high blood pressure during GA surgery could have more blood loss and a higher risk of requiring a blood transfusion. Borghi et al. [44] hypothesized that the anesthetic gas used in GA might inhibit erythrocyte production during the endogenous recovery of erythrocytes. Sessler et al. [45] theorized that the reduction in body temperature during GA may cause coagulation disorders and could increase the requirement for a blood transfusion. Our data showed no significant group differences between the type of anesthesia, in contrast with the literature; several negative outcomes are linked to GA in elderly patients undergoing hip surgery, such as hypotension, venous thrombosis, and pulmonary infection. However, in our cohort of patients, blood transfusion was not statistically correlated with GA.

Our results did not show a correlation between the ASA score and the transfusion rate. The literature findings on this theme are controversial. Dillon et al. [46] reported evidence that was comparable to that of our study, while other authors [17,47] included the ASA score as a transfusion risk factor. Comparable results were found for the CCI score. Both the ASA and the CCI scale, as well as postoperative complications, were assessed as mortality risks at 30 days and 1 year after hip surgery [48,49,50], but to the best of our knowledge, this is the first study to have investigated the CCI as a risk factor for blood transfusion in elderly patients with eFNF.

The surgery duration influenced the risk of blood transfusion in patients who were treated for eFNF (*p* < 0.05). This was in contrast with the results reported in the literature [35]. The correlation between the two parameters remains controversial.

Some authors investigated the hidden blood loss in eFNF as the main component of the total blood, finding this element as a result of the initial trauma rather than the surgery [51].

In our series, no open reductions were performed, but it can be assumed that when an open reduction is required, the risk of blood loss is higher. Patients with unstable eFNF exhibited a greater hemoglobin drop and hidden blood loss [52]. In a meta-analysis from 2017 [53], Ma et al. analyzed two-screw and one-screw IMN devices, and they demonstrated a significant difference between subgroups for surgery duration, but not for blood loss. Other authors also studied the differences [54] in outcomes in patients in terms of the types of nailing in eFNF (intermediate versus long cephalomedullary nailing), but they were similar in terms of the rates of blood transfusion, perioperative complications, malunion, and death.

The time interval from admission to surgery seems not to be related to an increased risk of having a blood transfusion in our cohort of eFNF patients. Similar results were found in the study by Kim et al.: if an appropriate transfusion strategy is adopted, surgery after 48 h does not seem to affect the rate of blood transfusion in elderly patients with intertrochanteric fractures, although the authors stated that the careful management of pre-operative hemoglobin in transfusion is necessary when surgery is delayed [55].

In contrast to these findings, a recent study [18] demonstrated a positive correlation between blood-transfusion units and the timing of surgery. The authors reported a lower blood-transfusion risk in patients who were treated within 24 h; however, more liberal criteria for blood-transfusion administration (Hb < 90 g/L with signs or symptoms of anemia) were chosen compared to our protocol [18].

Focusing on proximal-femur-fracture stability, an increased rate of preoperative transfusions was found among elderly patients with unstable intertrochanteric or subtrochanteric fractures submitted to surgery with an intramedullary nail with a delayed surgery of more than 24 h [56]. Our institution has a restrictive transfusion threshold (Hgb < 8 g/dL) that was rigorously followed, and a mean of 0.82 units of blood were transfused to our patients, which is analogous to subsequent equivalent protocols and minor compared to more liberal management practices [18]. A recent trial reported no significant differences between patients who were treated within 36 h and a cohort of groups who were treated after 36 h [37]. We found similar results in our study. However, hip-fracture surgery within 48 h of admission is associated with better outcomes and is moderately recommended by the AAOS [8], although limited evidence supports not delaying hip-fracture surgery for patients who are taking aspirin and/or clopidogrel [8]. One of the most widespread motivations for delaying surgery in patients who are affected by extracapsular femoral fractures is an ongoing anticoagulant treatment that is considered to cause extra blood loss due to impaired platelet function [35].

In our study, the one-year mortality rate was not found to be statistically different between the NBT and BT groups; different findings were reported by Greenhalgh et al. [25], who found that receiving a blood transfusion during admission for hip fracture carried an increased risk of one-year mortality of almost two-and-a-half times.

A study by Johnston [31] with a cohort of 1007 patients had similar results to our investigation. They found that blood transfusion was not associated with a change in one-year mortality or infection rates in proximal femur fractures in the elderly, even though there was a statistical increase in mortality at 120 days from surgery. Yombi et al. [29] concluded that low Hb on admission, age and RBC transfusions are significantly associated with one-year mortality (23.5%) after hip-fracture surgery, independently of comorbidity-associated factors.

A cohort study on an elderly population with hip fractures by Shokoohi [57] found that blood transfusions were not associated with changes in mortality, but only with an increased rate of postoperative infection.

In subjects with risk factors for blood transfusion, multidisciplinary management should be performed with adjunctive therapies [58], and supplementation with iron, vitamin B12, and folate should also be considered.

Some pilot studies [59,60] demonstrated the hematopoietic efficacy of intravenous iron in eFNF patients with a reduction in the mortality rate and length of hospital stay, although there is currently no clear consensus on the optimal method of managing perioperative anemia, and the effects on older people is too small and too late to affect erythropoiesis and transfusion rates.

A recent meta-analysis by Xing et al. [60] investigated the perioperative administration of tranexamic acid in geriatric-trauma patients with proximal-femur fractures undergoing IMN, demonstrating its efficacy in prophylaxis for perioperative hemostasis without increasing the incidence of postoperative complications.

One of the limitation of this study is its small sample size. However, a prospective controlled study will be performed, and more data will be available. Additionally, the intraoperative crystalloid volume and the hidden blood loss during the surgical procedure of IMN were not assessed. Finally, there are no standardized international blood-transfusion criteria. The strength of this study is the fact that there is no similar evidence in the literature derived from analyses of the risk factors of blood transfusion in these kinds of patients.

## 5. Conclusions

Patients with lower preoperative Hb levels and longer surgery times have a high risk of requiring blood transfusion, and they should followed up closely during the peri-operative period. In our study, BMI, time to surgery, ASA score, type of anesthesia, and CCI score were not found to be valid parameters for predicting the need for blood transfusion in elderly patients.

## Figures and Tables

**Figure 1 jfmk-08-00027-f001:**
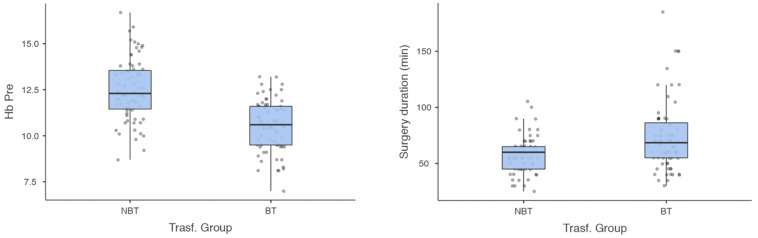
Pre-operative Hb levels and surgery duration (min) plots showing differences between NBT and BT groups (statistical significance of *p* < 0.001).

**Figure 2 jfmk-08-00027-f002:**
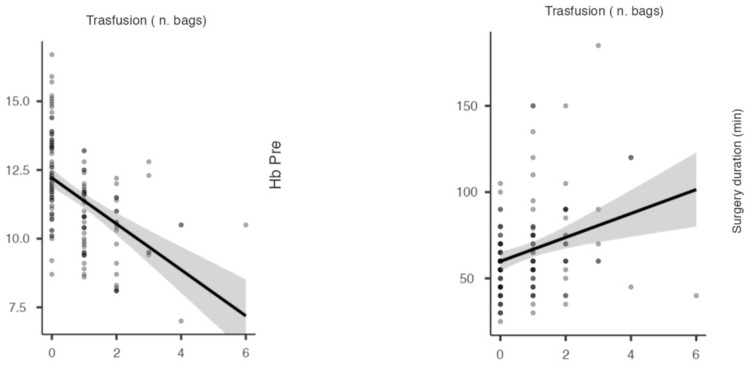
Pearson correlation graphs show the negative and positive relationships between the number of bags in transfusion, pre-operative Hb levels and surgery duration (min), respectively.

**Figure 3 jfmk-08-00027-f003:**
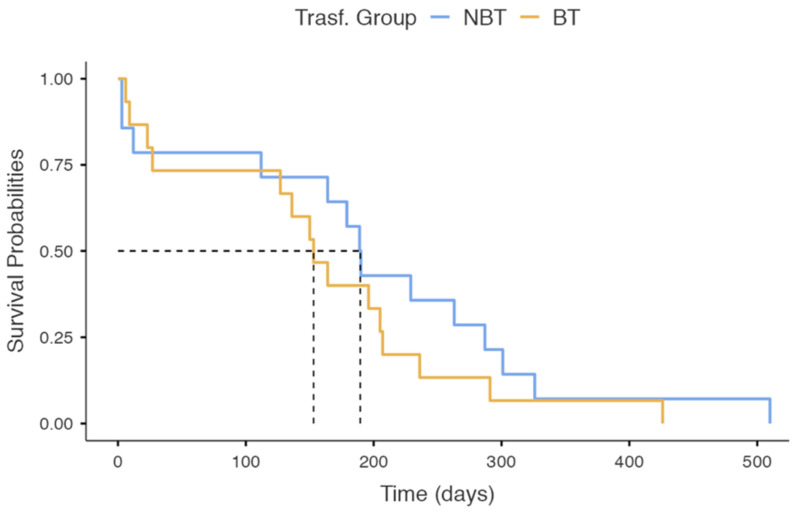
Kaplan–Meier survival plot between NBT and BT groups with a 1.5-year follow-up.

**Table 1 jfmk-08-00027-t001:** Patient-demographic information and study results.

	Patients (M:F)	Age (Range)	BMI	Hb	INR	Hospital Stay	Time To Surgery	Surgery Duration	Anesthesia (GA/CSEA)	ASA	CCI
**Group NBT**	71 (24:47)	80.2 ± 10.7 (65–94)	26.5 ± 5.9 (15.4–44.4)	12.5 ± 1.67 (8.7–16.7)	1.1 ± 0.12 (0.9–1.4)	10.8 ± 4 (5–24)	2.4 ± 1.6 (0–8)	57.8 ± 16.2 (25–105)	9/62	2.8 ± 0.6 (1–5)	5.21 ± 2.1 (1–10)
**Group BT**	72 (17:55)	82.9 ± 7.6 (65–99)	26.1 ± 5.1 (12.9–39.1)	10.6 ± 1.4 (7.0–13.2)	1.1 ± 0.16 (0.9–1.6)	11.4 ± 5.3 (5–39)	2.3 ± 2 (0–9)	73.3 ± 31.2 (30–185)	11/61	2.8 ± 0.5 (2–4)	5.78 ± 2.4 (2–12)
**NBT vs. BT**	0.17	0.07	0.65	<0.001	0.88	0.44	0.64	<0.001	0.65	0.81	0.19
**RR** **(95% C.I.)**	1.43 (0.8, 2.4)	1.52 (0.9, 2.5)	1.0 (0.77, 1.4)	1.03 (0.9, 1.7)	0.91 (0.49, 1.4)	1.16 (0.45, 2.8)	0.74 (0.5, 1.1)	1.37 (0.80, 2.3)	1.05 (0.9, 1.2)	0.96 (0.55, 1.6)	0.82 (0.62, 1.1)
**PCC** **(*p*)**	0.07 (0.64)	0.08 (0.33)	−0.033 (0.70)	−0.49 (<0.001)	−0.02 (0.89)	0.16 (0.06)	−0.07 (0.41)	0.28 (0.001)	0.11 (0.17)	0.02 (0.74)	0.16 (0.10)

NBT, no blood transfusion; BT, blood transfusion; BMI, body-mass index; Hb, hemoglobin; GA, general anesthesia; CSEA, combined spinal–epidural anesthesia; ASA, American Society of Anesthesiologists; CCI, Charlson Comorbidity Index.

## Data Availability

Study data are available from corresponding authors.

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
