# Peer review of "Blood-Transfusion Risk Factors after Intramedullary Nailing for Extracapsular Femoral Neck Fracture in Elderly Patients"

_jfmk, 2023, doi:10.3390/jfmk8010027_

Round 1
Reviewer 1 Report (Previous Reviewer 3)
The authors made the recommended corrections.Author Response
Thank you for your comments
Reviewer 2 Report (Previous Reviewer 2)
This revised version is much better.
They have changed their end point.
Conclusion- well written
Methods: well described but ideally should have more patients to Draw a definitive conclusion.
Results: well described
Discussion; Overall well discussed with evidence
Conclusion: tallies with results
I think this paper can be accepted with minor changes to language/English
Author Response
thank you for your comments
Reviewer 3 Report (Previous Reviewer 1)
Revised, no more comments.
Author Response
Thank you for your comments
This manuscript is a resubmission of an earlier submission. The following is a list of the peer review reports and author responses from that submission.
Round 1
Reviewer 1 Report
1. This study design is not a clinical trial, you may consider change the ‘trial’ to a proper term.
2. The author stated ‘The aim of this study was to identify and evaluate the perioperative risk factors that led to blood transfusion… ’ And the authors’ found that ‘Patients who had a lower preoperative Hb level had a high blood transfusion risk’. And the author stated that ‘Our Institution has a restrictive transfusion threshold (Hgb <8 g/dL) that was rigorously followed’ If this is already a requirement, why the authors conduct this research? Since if the patients’ baseline Hb level lower than <8 g/dL they will for sure get the transfusion.
3. The authors stated there was a table 1, however, there was no table 1 in the manuscript at all.
4. For the figure 4, the Kaplan-Meier survival curve, p values and comparison results should be given, however, except the in the statical analysis section, nothing were mentioned for this analysis.
Reviewer 2 Report
This paper needs some improvements. The numbers in the abstract and results do not tally. Also in the abstract it says GA increases the risk of blood transfusion but in the discussion section it says GA does not increase the risk of blood transfusion.
Abstract - The abstract numbers must tally with the results section.
Introduction - Introduction is okay.
Methods - Methods is okay. Maybe should include a longer duration to get a larger samples size.
Results - Results numbers do not tally up with abstract numbers. Please have a look at this.
Discussion - Discussion is okay. So does GA increase the risk of transfusion? Please have a look at your abstract and your statement in the discussion section. Otherwise discussion is okay.
Conclusion - Conclusion is fine.
English language needs improving, Some spelling and grammar mistakes.
Needs major revision.
Reviewer 3 Report
Interesting article on a topic not yet fully explored. My comments: Introduction : Please write if there have been similar studies assessing the need for blood Transfusion after Intramedullary Nailing for Femoral Neck Fracture. Please change ...Hip fracture…. On Femoral Neck Fracture or intertrochanteric fractures, as appropriately named. Discussion Please change ...Hip fracture…. On Femoral Neck Fracture or intertrochanteric fractures, as appropriately named. Please write that there is a high mortality rate after Femoral Neck Fracture and identifying blood transfusion risk factors can theoretically reduce mortality.